# Correlations Based on Numerical Validation of Oscillating Flow Regenerator

**Kuruchanvalasu Jambulingam Bharanitharan** [1] **, Sundararaj Senthilkumar** [1] **, Kuan-Lin Chen** [2] **, Kuan-Yu Luo** [2] **and Shung-Wen Kang** [2,*]

1 Department of Aerospace Engineering, SRM Institute of Science and Technology, Kattankulathur, Chennai 603203, India; bj7005@srmist.edu.in (K.J.B.); senthils7@srmist.edu.in (S.S.)
2 Department of Mechanical and Electro-Mechanical Engineering, Tamkang University, New Taipei City 25137, Taiwan; chy130367@gmail.com (K.-L.C.); luoguanyu0829@gmail.com (K.-Y.L.)
* Correspondence: swkang@mail.tku.edu.tw; Tel.: +886-2-2621-5656 (ext. 3279)

**Abstract:** Stirling regenerator is one of the emerging heat exchanger systems in the area of cryogenic cooling. Many kinds of research have been conducted to study the efficiency of Stirling regenerators. Therefore, the principles and related knowledge of Stirling refrigerators must be thoroughly understood to design a regenerator with excellent performance for low-temperature and cryogenic engineering applications. In this study, an experimental setup is developed to estimate the pressure drop of the oscillating flow through two different wire-mesh regenerators, namely, 200 mesh and 300 mesh, for various operating frequencies ranging from 3 (200 RPM) to 10 Hz (600 RPM). Transient, axisymmetric, incompressible, and laminar flow governing equations are solved numerically, and source terms are added in the governing equations with the help of the porous media model and the Ergun semiempirical correlation, assuming that the wire meshes are cylindrical particles arranged uniformly. Simulation results show that the numerical predictions of temporal pressure variation are in reasonably good agreement with those of experimental findings. It is also found that the Ergun correlation works more accurately for higher flow rate conditions.

**Keywords:** Stirling regenerator; porous media; oscillating flow; wire-mesh regenerator; pressure drop characteristics

## 1. Introduction

Stirling engine is a small-scale regenerator [1]. However, compared with the usual refrigeration technique in the Stirling engine, the working fluid can be any natural gas. Hence, the disadvantages of refrigerant were reduced.

An experimental setup was developed by Hsu et al. [2] to estimate the pressure drop and velocity of both steady and oscillating flow regenerators. The study reveals that the system showed a quasi-steady-state behavior under 4 Hz (maximum operating frequency). It is found that both steady and unsteady correlations were in agreement with each other. A numerical study was performed using the Brinkman–Forchheimer extended model to model the momentum in a porous region by Guo et al. [3] to study the effect of hydrodynamic characteristics for a partially filled porous regenerator. The study was performed with various porous layers with different thicknesses. The conclusion stated that the effectiveness of thermal diffusivity on the thickness of the porous layer is reduced with an increase in the Darcy number. A new flow model was developed to overcome the inaccuracy of the convention flow models for pulsating flow regenerators. An additional term known as breathing factor was introduced in the mass balance equation to improve the solution accuracy [4]. An experimental study was performed by Choi et al. [5] to study the oscillating flow characteristics of a wire matrix regenerator. As a result, two numerical correlations were developed. It is noted that the friction factor and phase angle of pressure drop are a function of the Reynolds number, the Valensi number, and the ratio of the flow

domain, respectively. Later, the experiment was performed with helium as working fluid, and the same relations were studied under room temperature conditions. It was found that the friction factor model obtained for room temperature condition was inaccurate for the model with cryogenic conditions at the cold end [6].

A pressure drop study on an oscillating flow regenerator with a metal foam structure shows that the pressure drop increases with an increase in velocity. The flow condition is mostly sinusoidal due to the to-and-fro motion of fluid inside the regenerator. It is also stated that pressure drop is less for metal foam with an open-cell structure compared with the other materials used in the study. The Reynolds number based on oscillating flow characteristics was found to be the most significant parameter that affects the flow pressure drop [7]. A comparative study on 3 He and 4 He (isotopes of helium) was performed by Radebaugh et al. [8], and regenerator performance was calculated.

Kim and Ghiaasiaan [9] performed a numerical study on a pulsating laminar regenerator. Within the range of the study, it was concluded that the cycle average permeability coefficient was strongly dependent on the porosity of the regenerator. The Forchheimer inertial coefficient was dependent of the pulsating frequency of the flow. Based on experimental results, an anisotropic equation was derived from calculating the permeability and the inertial coefficient for a wire screen mesh regenerator by Tao et al. [10]. A computational study was performed by Teitel [11] to simulate the flow through a woven screen mesh using available correlation data, and the permeability and inertial coefficient were obtained from the wind tunnel testing of Miguel [12], and the obtained numerical results were validated well with the experimental data.

A 2D staggered cylindrical array model was designed to approach the practical wire screen or packed regenerator to reduce the computational resources and time. The model was able to predict the experimental results more accurately, and it is also stated that the model was capable of predicting the results beyond the scope of the experiments [13]. A numerical model was developed with an actual woven screen structure to derive a correlation equation for estimating the pressure loss in a woven screen or stacked regenerator. It was found that the estimated numerical correlation best fits with two parameters of the Ergun correlation [14–16]. An experimental analysis was performed with a packed bed regenerator to study the thermohydraulic characteristics of an oscillating flow regenerator. The conclusion stated that the matrix with spheres of a small diameter and a high flow rate had the highest pressure. In contrast, the matrix with a larger diameter and a lower flow rate had a comparatively lower pressure drop [17,18]. Boroujerdi and Esmaeili [19] performed a numerical analysis of the wire screen regenerator to estimate the frictional losses and heat transfer. The viscous resistance and inertial resistance were estimated using geometric and material properties, such as porosity, permeability, and hydraulic diameter. The developed correlation reveals that the pressure drop and heat transfer increase as the wire diameter decreases. Sadrameli [20] reviewed the mathematical model used for the computation simulation of Stirling regenerators. The review proves the importance of choosing a proper mathematical model to predict accurate results, especially to measure outlet temperature and regenerator efficiency. A numerical simulation of a small-scale miniature pulse tube regenerator was performed to study the effect of operating conditions by Poshtkouhian et al. [21]. The study concludes that increasing the filling pressure and frequency of operation improves the heat transfer characteristics of the regenerator. Alfarawi et al. [22] performed a three-dimensional computational study to predict the potentiality of the application of a miniature Stirling engine for real-time applications. The study concludes that the regenerator of a 0.5 mm channel used in the study has a good potential to maximize the engine power. Peng et al. [23] performed an experimental study to compare three different types of regenerators, such as foam, packed bed, and wire type. Out of the three models used in the study, the foam type is more efficient because of its lower pressure drop. It is found that there is a threshold value for a specific area for particle and wire screen. Under this threshold region, both particle and wire screen regenerators have higher effectiveness than foam-type regenerators, but beyond the threshold region,

a foam type-regenerator becomes superior. Sowale and Odofin's [24] study on losses in free piston shows that the thermal efficiency of the regenerator is highly affected by the performance efficiency of the regenerator. From the study, it is noticed that the heat transfer is enhanced with an increase in porosity; on the contrary, the hydrodynamic performance is reduced. A CFD analysis was performed to estimate the pressure drop of the oscillating flow regenerator [25]. A numerical analysis was performed with a simplified 2D circular array assumed regenerator model [26]. In steady flow conditions, the model validated the experiment more accurately, whereas in transient conditions, the deviation was estimated to be 7%. A 2D asymmetric free-piston Stirling engine model was developed to investigate the working mechanism of the regenerator using CFD [27]. The deviation between experimental and numerical results was found to be 27%. This deviation is due to the negligence of internal losses in the engine, heat transfer losses, and losses due to contact resistance. A friction factor correlation was developed using a full-scale 3D analysis of a β-type Stirling regenerator in Ansys Fluent with a porous media model [28]. The permeability and inertial coefficients were obtained using curve fitting of steady-state results.

Jeong et al. [29] performed an experimental and numerical simulation to understand the characteristics of the pulsating regenerator. A numerical analysis for oscillating flow was performed using the steady-state friction factor. The results showed a good validation in pressure drop for the model with lower frequency, but as the frequency increased, there was a decrement in the amplitude of pressure drop.

A Matlab code was developed to test different numerical algorithms for an ideal closed-cycle regenerative cryocooler [30]. The parametric study reveals that the regenerator with a higher length-to-wire-diameter ratio is comparatively inefficient compared with the model with a lower length-to-diameter ratio. Table 1 shows various numerical correlations reported earlier by other authors.

Based on the literature study, it is found that, normally, the correlations for the friction factor of a wire-mesh regenerator have been developed based on the huge experimental data, which are too expensive as compared with numerical simulations. Then the inertia and viscous resistance parameters were obtained from those experimental data and were used for the numerical studies for a wire-mesh regenerator. Hence, in the present work, an attempt is made to formulate the correlations based on numerical results obtained with a uniform cylindrical particle assumption of wire-mesh regenerators after thoroughly validating with those of experimental findings. This is envisaged in two steps. First, an experimental setup is developed with two different wire-screen uniformly arranged mesh regenerators to estimate the pressure drop characteristics at a 6 bar operating pressure condition for various oscillating frequencies ranging from 200 to 600 RPM. Second, a numerical system is developed using a dynamic mesh, and the resistance parameters are obtained using a semiempirical Ergun correlation. The Ergun correlation is one of the standard empirical relations used in particle-based porous flows. Here, it is assumed that the cylindrical wires in a mesh structure are divided into microscale uniform cylindrical particles. Using this assumption, resistance parameters are calculated using the semiempirical Ergun correlation, and later, it is used for a numerical study to validate with the experimental results. By performing this, a single correlation for a fixed mesh can be used for various studies, considering that the mesh structure is uniform. This study mainly focuses on the verification of this assumption using combined experiment and CFD techniques and formulates the correlations for the same.

**Table 1.** Friction factor correlation obtained in various experiments.

| Author | Regenerator Matrix | Friction Factor | Validity | Comment |
|---|---|---|---|---|
| Miyabe et al., 1982 [31] | Woven screen | $f = \frac{33.6}{Re} + 0.0337$ | $0.586 < \epsilon < 0.840$<br>$5 < Re < 1000$<br>Fluid—$N_2$ | Only applicable for steady flow |
| Tanaka et al., 1990 [32] | Woven screen, sponge metal, and sintered metal | $f_h = \frac{175}{Re} + 1.6$ | $0.645 < \epsilon < 0.729$<br>Fluid—Air | $f_{osc} > f_{st}$ |
| Zhao and Cheng 1996 [33] | Woven screen | $f = \frac{1}{(A_0)Dh}\left[\frac{247.3}{Re_\omega} + 1003.6\right]$ | $0.01 < Re_\omega < 0.13$<br>$0.602 < \epsilon < 0.662$<br>$Re_\omega = \frac{\omega D_h^2}{\gamma}$<br>Fluid—Air | The friction factor for an oscillating flow is 4 to 5 times higher than that for a steady flow |
| Gedeon and Wood 1996 [34] | Woven screen | $f = \frac{129}{Re} + 2.91Re^{-0.103}$ | $0.6232 < \epsilon < 0.7102$<br>Fluid—$N_2$, He | No difference if $V_a < 20$ |
| Pamuk and Mustafa 2011 [35] | Packed balls | $f_{max} = \frac{3083998}{Re_{max}} + 1882$<br>$f_{max} = \frac{532936}{Re_{max}} + 612.1$ | $\epsilon = 0.369$<br>$\epsilon = 0.3912$ | Increasing porosity reduces the maximum friction factor |
| Xiao et al., 2017 [36] | Woven screen | $f = \frac{134}{Re_h} + 5.44Re_h^{-0.103}$ | $0.665 < \epsilon < 0.78$<br>$Re_h = \frac{uD_h^2}{\gamma}$ | Only applicable for this experiment range |
| Mingjiang et al., 2017 [37] | Wire screen (100 mesh to 400 mesh) | $\overline{f} = \frac{1}{A_o^*}\left(\frac{D}{Re_{\omega(dh)}} + E\right)$<br>$\overline{f}$ denotes the cycle averaged friction factor | $0.00309 < Re_{\omega(dh)} < 0.220$<br>$3150 < A_o^* < 13100$ | $A_{(dh)} = \frac{x_{max}}{d_h}$<br>$A_o^* = \delta * A_{(dh)}\frac{\rho_{cycle}}{\rho_{mean}}$ |

## 2. Experiment Design and Setup

The regenerator is the most crucial part of the Stirling refrigerator because it acts as a heat exchanger in the system. The regenerator absorbs heat from the working gas when the gas flows from the hot end to the cold end and releases heat to fluid during the reverse cycle. The experimental test rig is shown in Figure 1. In this experimental device, the motor drives the crankshaft, and the crankshaft guides the bellows to compress and expand the gas so that it is an oscillatory bidirection flow system.

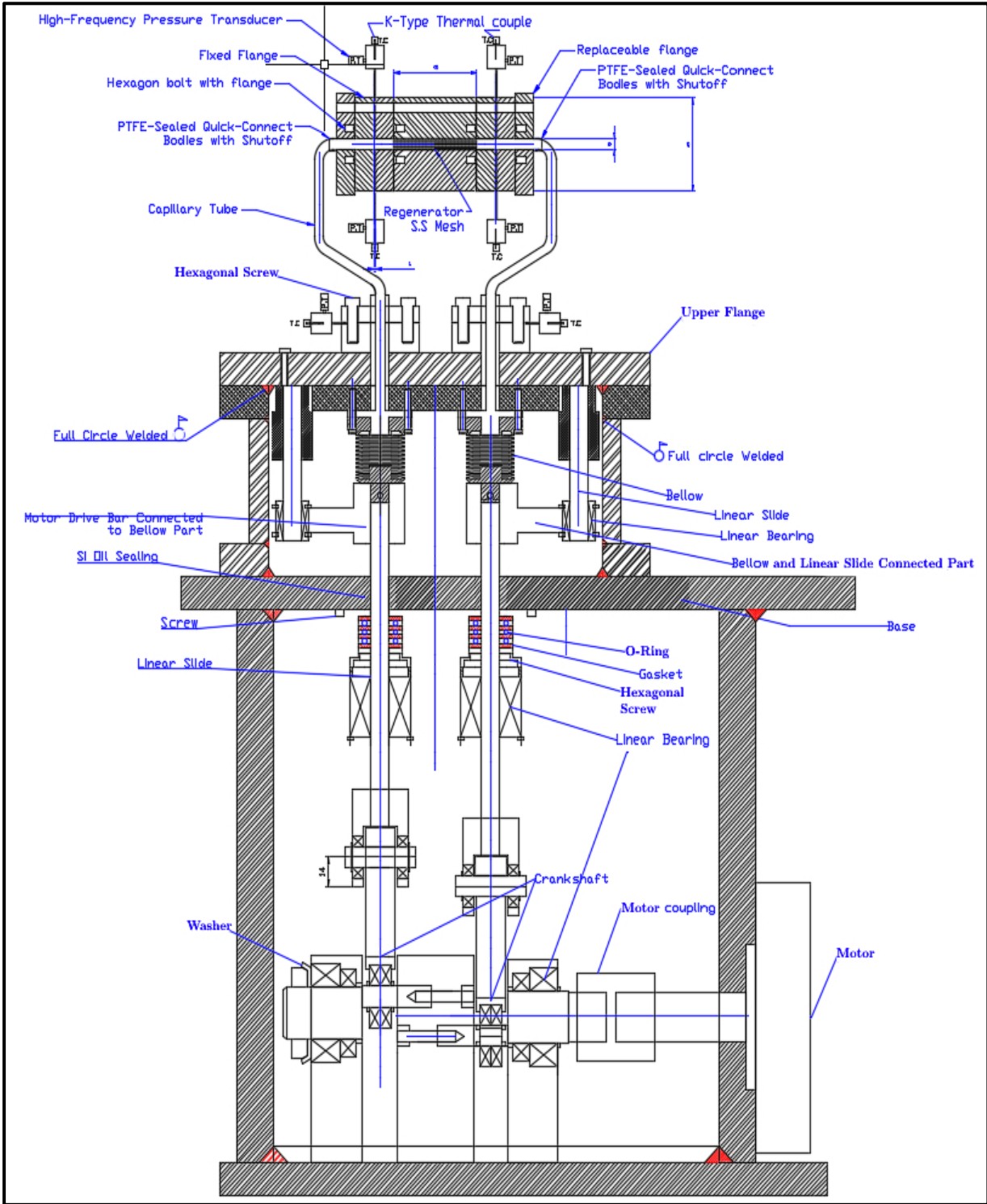

**Figure 1.** Experimental test rig.

### 2.1. Working Fluids

The three common working fluids used in the Stirling engine are air, hydrogen, and helium. The most recommended working fluid is helium (3 He and 4 He). Helium is preferred because of its high thermal conductivity, low condensation point (4K), ease of obtaining, and low risk. In this study, highly pure helium is used as a working fluid. Helium has low viscosity, and it can flow through the regenerator easily. This study utilizes the fluid at an ambient temperature condition as this study concentrates more on the hydrodynamics of the regenerator.

### 2.2. Regenerator

The heat exchanging media (matrix) is made of a light felt like a mass of fine wire stacked in a well-insulated tube, as shown in Figure 2. It shows the stacking of wire-mesh screens inside the regenerator. The fine wire mesh used in the regenerator is commonly obtained in the form of a woven screen at various wire sizes, weave structures, mesh densities, and materials. The wire screen is made of stainless steel, as it is easy to obtain, low cost, easy to process. In this experiment, two regenerators were selected, 200 mesh and 300 mesh. The prefix number used for mesh denotes the number of screens used in the regenerator. The length of the three regenerators is $L_r$ = 45 mm, and the inner diameter is $d_r$ = 5 mm. The wire diameter and porosity of the regenerator are shown in Table 2. Figure 3 shows the location of pressure ports and regenerator fix block.

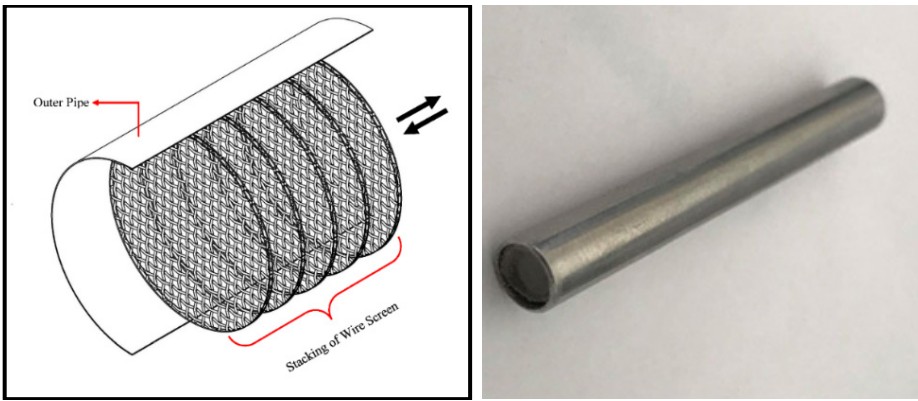

**Figure 2.** Stacking of a wire screen mesh inside the regenerator.

**Table 2.** Regenerator parameters.

| No. of Mesh Screens | Wire Diameter (μm) | Wire Center Pitch (μm) | Porosity | Cross-Section Area ($10^{-5}$ m$^2$) | Hydraulic Diameter (μm) | Outer Diameter (m) | Regenerator Length (m) |
|---|---|---|---|---|---|---|---|
| 200 | 39.07 | 125.031 | 0.7489 | 1.3927 | 73.37 | 0.005 | 0.045 |
| 300 | 40.70 | 85.339 | 0.5849 | 1.4598 | 78.92 | 0.005 | 0.045 |

### 2.3. Bellow

In this experiment, the function of the bellow is to compress and expand the working gas and withstand the pressure difference between the pressure chamber and the crankcase for a long time. The primary purpose of replacing the piston with a bellow in this experimental setup is to reduce the pressure loss due to leakage. The material used for the bellow is AMTM350. The material is chromium–nickel–molybdenum precipitation hardening stainless steel, as shown in Figure 4. The diameter of the bellow is 19 mm, and it has a stroke length of 16 mm. When the bellow is at the top dead center, the maximum volume is calculated to be 11.9 cc, and the minimum volume when the bellows are at the bottom dead center is estimated to be 7.26 cc. The sweeping volume $V_{swept}$ is calculated through AutoCAD and found to be 4.64 cc, and the dead zone volume $V_{dead}$ is 0.13 cc.

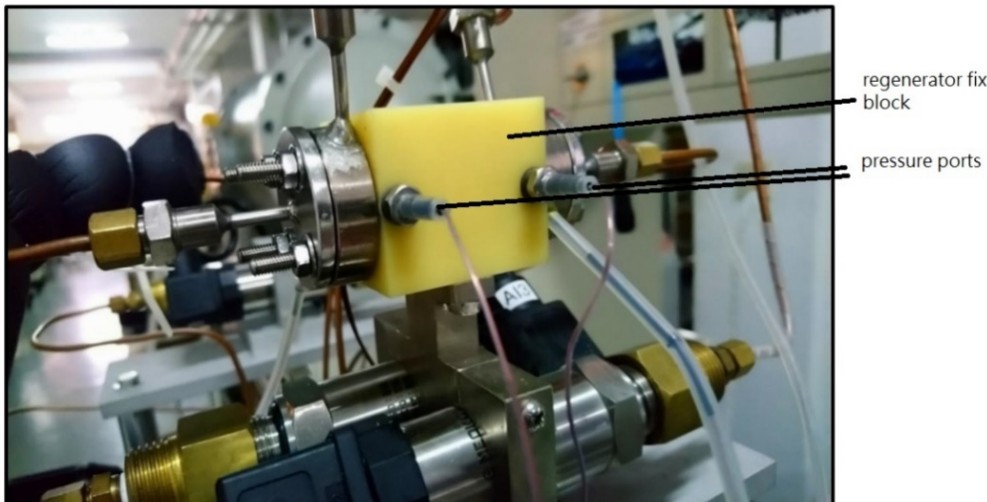

**Figure 3.** Regenerator fix block.

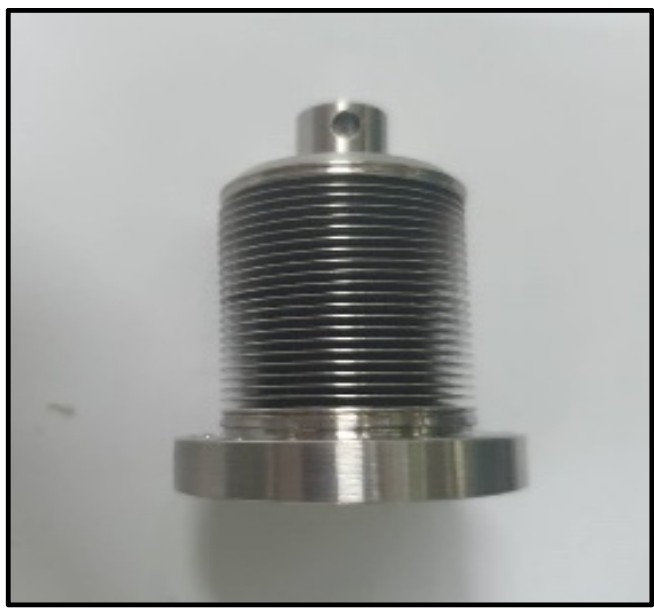

**Figure 4.** Bellow.

*2.4. Pressurization*

Pressurized gas is usually used to improve performance. The higher the pressure, the higher the performance rate, and adversely, it requires a higher torque to run the motor, which leads to higher power consumption. The degree of pressurization needs to be considered based on the motor's limit, the strength of the material, and the system's internal pressure. The test section of the regenerator in this experiment is filled with 6 bar helium gas, and the balance chamber is filled with air to 6 bars so that the welded bellows in the balance chamber can work smoothly.

*2.5. Measuring Technique*

The pressure gradient at the inlet and outlet of the regenerator is measured by pressure transducers from Meokon Sensor Technology, Shanghai, China, and its model is MD-HF, shown in Figure 5. The pressure range is 0–3 MPa with an accuracy of 0.5%. The highest response frequency can reach 200 kHz. The flow data are extracted through the data acquisition module, and the motor speed can be found using the motor servo. The fluid oscillates from the hot end to the cold end. In order to measure the temperature

difference inside the regenerator, a k-type thermocouple is used. The cold-end part of the regenerator uses air as natural convection for heat dissipation. The pipe is partially covered with insulation material to prevent excessive heat loss. A fixed block is used to hold the regenerator. In order to know the errors occurring in the experiment, a leak test is performed before the start of experiments, and it is found that the leakage mainly includes small cracks through welding parts and uneven screw locking. The leakages are found using a helium leak detector (Model Phoenix L300I, Leybold GmbH, Cologne, Germany). It consists of a probe used to check the helium leakage maximum up to $10^{-10}$ bar. There is a display that shows the graph. If there is any leakage, then it shows deflection, and the buzzer sound will instantly change to a high pitch. Once the causes of the leakage are known, the experiments are performed by preventing the causes, leading to leakage.

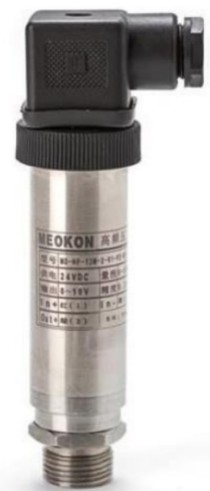

**Figure 5.** Pressure transducer.

*2.6. Error Analysis*

To understand the temporal error propagation in the obtained experimental study, a statistical study is performed. The standard deviation of the mean is obtained for a multiple set of experiments performed under the same operating condition up to a flow time of 60 s. With a 95% confidence level using the *t*-test, uncertainties are obtained using Equation (1):

$$u_i = \pm K \overline{\sigma} \tag{1}$$

where K and $\overline{\sigma}$ are *t*-value (2.57) and standard deviation of the mean. From a thorough analysis for different operating conditions, the uncertainties are found to vary between $\pm 0.221$ and $\pm 0.55$ bar.

**3. Numerical Modelling**

*3.1. Geometry and Meshing*

To simplify the case and reduce computational time, the flow can be assumed to be 2D axisymmetric as the flow variations are more significant in both axial and radial directions only. The 2D model for CFD simulation is modeled using an Ansys Design Modeler, and a schematic diagram of the computational geometry is shown in Figure 6. The design consists of three main parts: hot- and cold-end bellow, regenerator, and pipe for flow to achieve entrance length. A uniform quadrilateral mesh of $1.52 \times 10^5$ elements has been developed, and it is shown in Figure 7, which is a zoom of the mesh structure for a clear visibility of the mesh. After a thorough grid-independent test, it is found that a further increment in the number of grids does not have any effect on the results. Figure 8 shows the boundary conditions used in the numerical study. No slip boundary condition is used for all the walls of the bellows, regenerator, and pipe. The oscillatory velocity boundary conditions are realized at the hot end and cold-end bellows through a user-defined function

where the axial velocity, $v_z$, is specified as $v_z = \left(\frac{L}{2}\right)\sin(\omega t)$, where L is the swept length and $\omega$ is the angular frequency of the bellow.

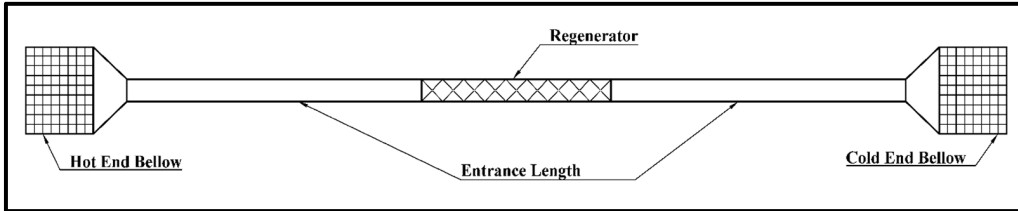

**Figure 6.** 2D Computational domain of the problem.

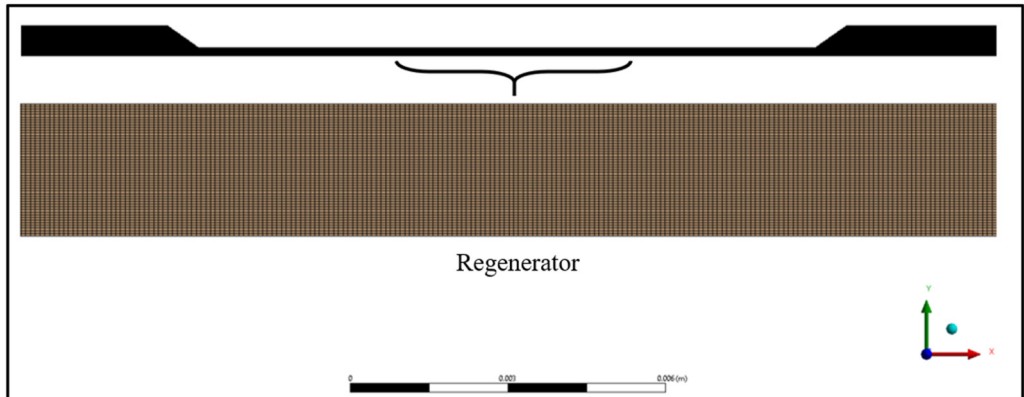

**Figure 7.** Meshing of the numerical model.

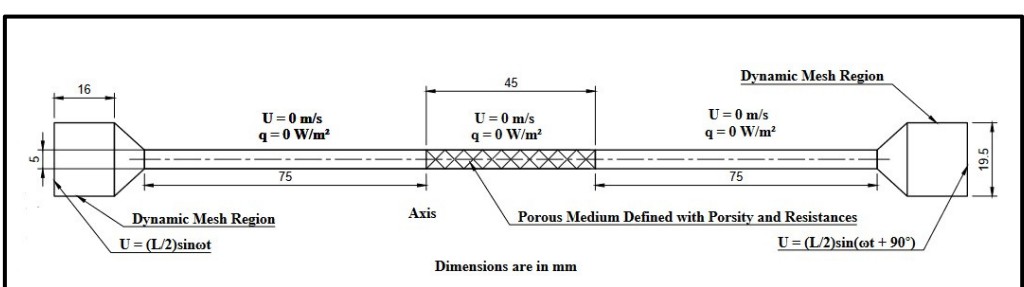

**Figure 8.** Boundary conditions used for numerical simulations.

In Figure 8, the left-most domain and right-most domain indicate the hot- and cold-section bellows, respectively. They are modelled using the dynamic mesh model by defining the bellow motion. A phase difference of 90° is maintained in between two bellows. U and q in Figure 8 indicate the velocity magnitude and heat flux. A 75 mm pipe is defined in between the regenerator and the dynamic mesh region, allowing the flow to develop.

### 3.2. Porous Definition in Numerical Study

The regenerator section shown in Figure 6 is modelled using a porous zone in Fluent (Ansys, Inc., Pittsburgh, PA, USA) instead of creating an actual wire-mesh geometry. In order to assume a fluid cell zone that is porous, Fluent requires the following three parameters: viscous resistance, inertial resistance, and porosity. Based on the porosity, the regenerator zone is assumed as a packed structure considering that the wire mesh is uniformly spaced and screens are closely arranged. The viscous resistance ($R_v$) and inertial resistance ($R_i$) are calculated using the Ergun and Blake–Kozeny correlation [38,39] shown in Equations (2) and (3). Table 3 shows the calculated values of $R_v$ and $R_i$ using the above-mentioned assumptions and considering that the flow in the porous zone is laminar.

Inside the porous medium, the flow in the radial direction is restricted by providing a large value of resistance:

$$R_i = 3.5 \times \frac{(1-\varepsilon)}{\varphi \times D_w \times \varepsilon^3} \tag{2}$$

$$R_v = 150 \times \frac{(1-\varepsilon)^2}{\varphi^2 \times (D_w)^2 \times \varepsilon^3} \tag{3}$$

where, $\varepsilon$ is the porosity, $\varphi$ is the sphericity, and $D_w$ is the diameter of the wire.

**Table 3.** Inertial resistance and viscous resistance.

| Mesh | $D_w$ in μm | $\varepsilon$ | $\varphi$ | $R_v$ in m$^{-2}$ | $R_i$ in m$^{-1}$ |
|------|------|------|------|------|------|
| 200 | 39.07 | 0.7489 | 0.8218 | $2.345 \times 10^{10}$ | $6.835 \times 10^{04}$ |
| 300 | 40.70 | 0.5849 | 0.8727 | $1.022 \times 10^{11}$ | $2.043 \times 10^{05}$ |

*3.3. Governing Equations*

The flow is assumed to be 2D axisymmetric, unsteady, and incompressible as the Mach number is less than 0.3, and also, isothermal conditions are used in this study as the present study aims to predict only the hydrodynamic characteristics of the regenerator. Hence, the two-dimensional unsteady governing Equations (4)–(6) in the polar coordinate system are considered. In the momentum Equations (5) and (6), source terms are added in order to incorporate the effects of porosity. Helium is used as a working fluid at an operating pressure of 6 bars. The oscillatory motion is achieved using a dynamic mesh model. A user-defined function has been developed to define the bellow motion using Ansys C programming libraries. The velocity of the bellow is defined in Equations (7) and (8). A phase difference of 90° is used between two bellows so that one bellow will undergo compression when the other one is undergoing an expansion phase. The flow-governing equations, including continuity and momentum equations, are as follows [40,41]:

Continuity

$$\frac{\partial v_r}{\partial r} + \frac{\partial v_z}{\partial z} = 0 \tag{4}$$

R—momentum

$$\frac{1}{\varepsilon}\frac{\partial v_r}{\partial t} + \frac{1}{\varepsilon^2}\left(v_r\frac{\partial v_r}{\partial r} + v_z\frac{\partial v_r}{\partial z}\right) = -\frac{1}{\rho}\frac{\partial P}{\partial r} + \frac{\nu}{\varepsilon}\left(\frac{\partial^2 v_r}{\partial r^2} + \frac{\partial^2 v_r}{\partial z^2} - \frac{v_r}{r^2}\right) - \frac{\nu v_r}{k} - \frac{F v_r^2}{\sqrt{k}} \tag{5}$$

z—momentum

$$\frac{1}{\varepsilon}\frac{\partial v_z}{\partial t} + \frac{1}{\varepsilon^2}\left(v_r\frac{\partial v_z}{\partial r} + v_z\frac{\partial v_z}{\partial z}\right) = -\frac{1}{\rho}\frac{\partial P}{\partial z} + \frac{\nu}{\varepsilon}\left(\frac{\partial^2 v_z}{\partial r^2} + \frac{\partial^2 v_z}{\partial z^2}\right) - \frac{\nu v_z}{k} - \frac{F v_z^2}{\sqrt{k}} \tag{6}$$

where $v_r$ and $v_z$ are the velocity components in radial velocity and axial directions, respectively; P, the pressure; $\rho$, the density; $\nu$, the kinematic viscosity; $\varepsilon$, the porosity; t, the time; r and z, the radial and axial coordinate systems; k, the permeability; and F, the Forchheimer parameter. The Forchheimer parameter and inverse permeability are obtained using Equations (2) and (3), respectively.

Velocity on bellow

$$v_{p1} = \frac{L_s}{2}\sin(\omega t) \tag{7}$$

$$v_{p2} = \frac{L_s}{2}\sin\left(\omega t + \frac{\pi}{2}\right) \tag{8}$$

*3.4. Numerical Discretization*

The governing equations are discretized using the finite volume method using Fluent. Pressure velocity coupling is achieved using the SIMPLEC [42] algorithm. The pressure equation is discretized using a second-order scheme. The convective terms in the momentum equations are discretized using a second-order upwind scheme, and diffusion terms are discretized with a second-order scheme. A first-order implicit scheme is used for discretizing the temporal term. The time step size ($\Delta t$) is chosen as $3 \times 10^{-5}$ s. The under-

relaxation values used for pressure and momentum equations are 0.3 and 0.7, respectively. The convergence criteria are chosen as $10^{-6}$ for continuity and momentum equations. The flow is initialized with an operating pressure of 6 bar. The simulations are performed for a flow time of 1 s.

## 4. Results and Discussion

The pressure drops are calculated using both the CFD and experimental methods for two different mesh sizes of 200 and 300 at a range of angular velocity of a bellow from 200 to 700 RPM with an increment of 100 RPM. They are also repeated for both hot- and cold-end sides as the flow direction changes with time due to bidirectional motion of the bellow with a phase difference of 90°. The numerically estimated pressure drops are found to be in agreement with the experimental results. The characteristics of pressure variation along the axial direction in different cycles are studied. A friction factor correlation is also defined using two coefficients Ergun equation with the numerically obtained maximum velocity.

### 4.1. Hot- and Cold-Section Pressure Characteristics

Figures 9–13 show the hot- and cold-section pressure characteristics of both experiment and CFD data for a 300 mesh regenerator, and Figures 14–18 show the pressure characteristics of a 200 mesh regenerator. In Figure 9, for the 300 mesh regenerator, it can be inferred that the cold-end pressure has a specific time lag with the experimental results. The numerically estimated maximum and minimum pressures deviate by 0.1% and 4.4% from experimental results along with this time lag. As the operating frequency increases from Figures 10–13, the time lag between the experimental and numerical results vanishes, and also, the error in deviation of the maximum pressure is reduced to 2%. It can also be inferred that as the flow time increases, the error rises in the numerical result.

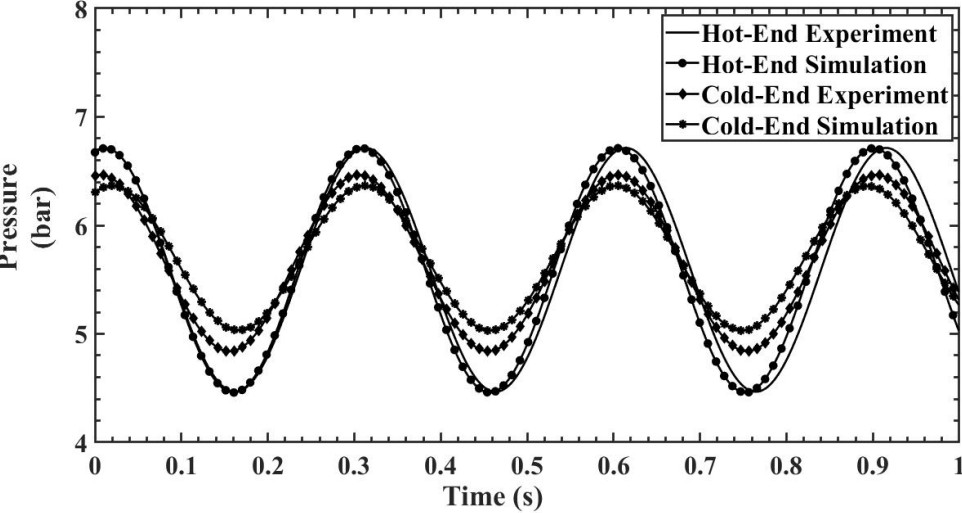

**Figure 9.** Hot- and cold-section pressure characteristics of a 300 mesh regenerator at 200 RPM.

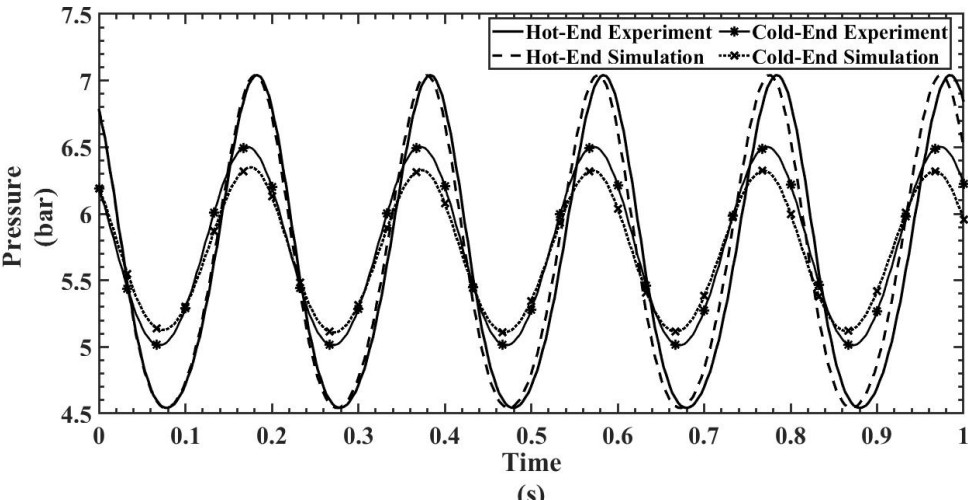

**Figure 10.** Hot- and cold-section pressure characteristics of a 300 mesh regenerator at 300 RPM.

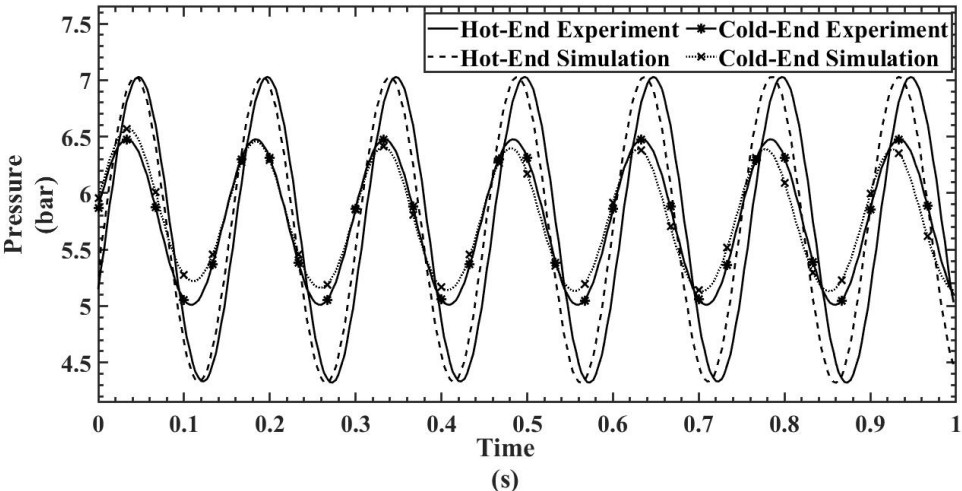

**Figure 11.** Hot- and cold-section pressure characteristics of a 300 mesh regenerator at 400 RPM.

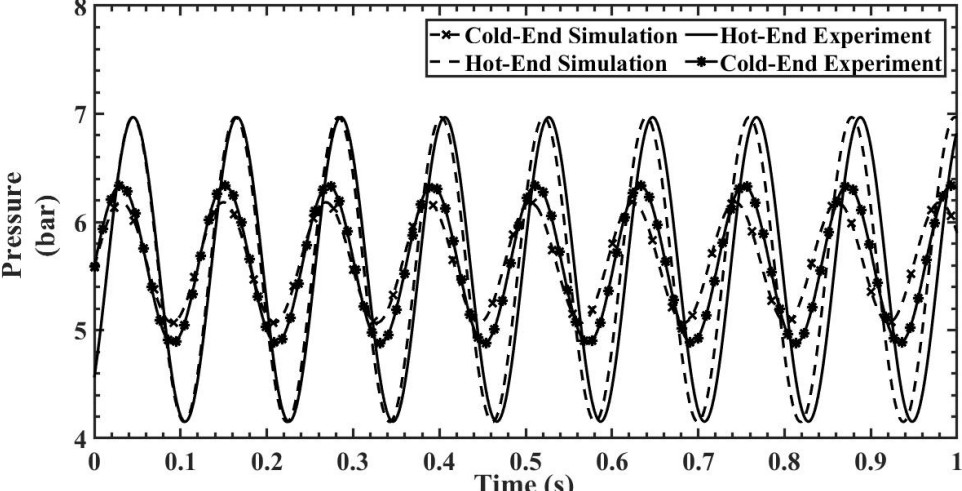

**Figure 12.** Hot- and cold-section pressure characteristics of a 300 mesh regenerator at 500 RPM.

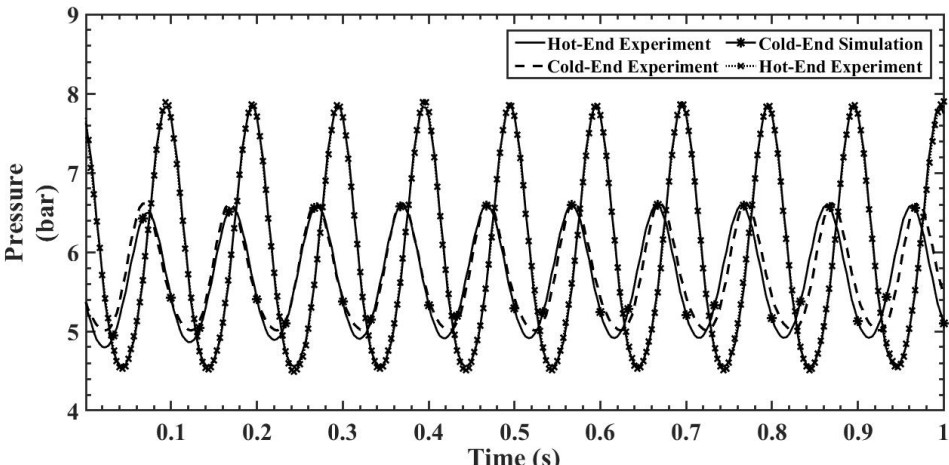

**Figure 13.** Hot- and cold-section pressure characteristics of a 300 mesh regenerator at 600 RPM.

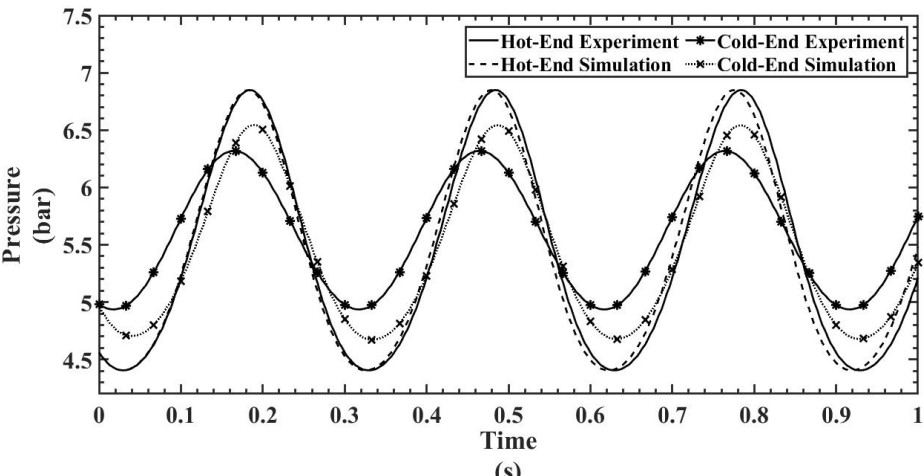

**Figure 14.** Hot- and cold-section pressure characteristics of a 200 mesh regenerator at 200 RPM.

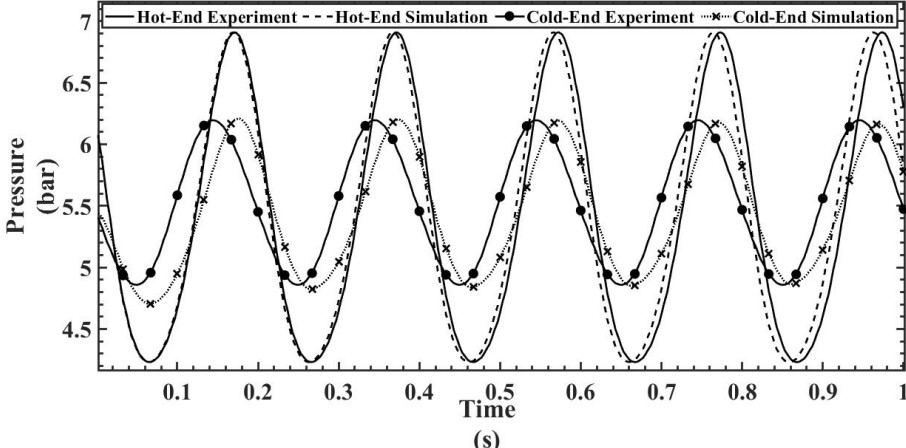

**Figure 15.** Hot- and cold-section pressure characteristics of a 200 mesh regenerator at 300 RPM.

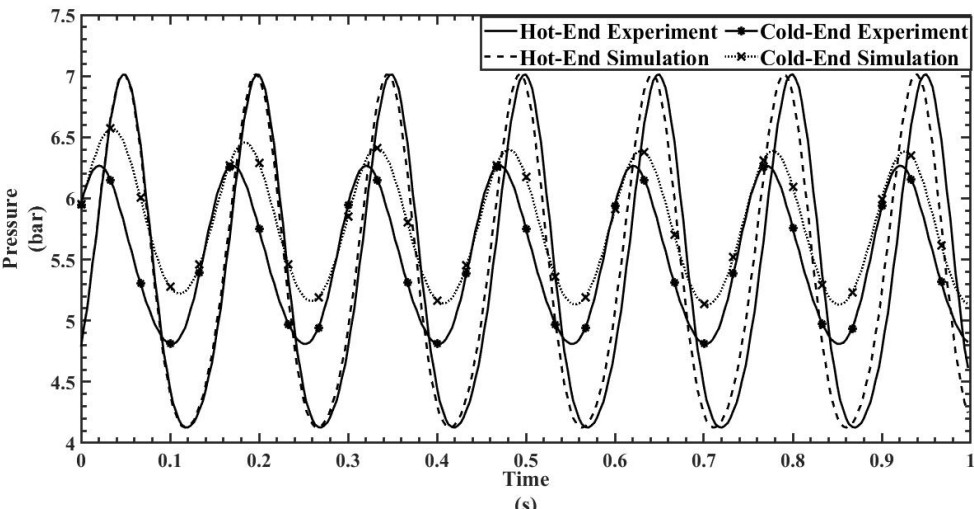

**Figure 16.** Hot- and cold-section pressure characteristics of a 200 mesh regenerator at 400 RPM.

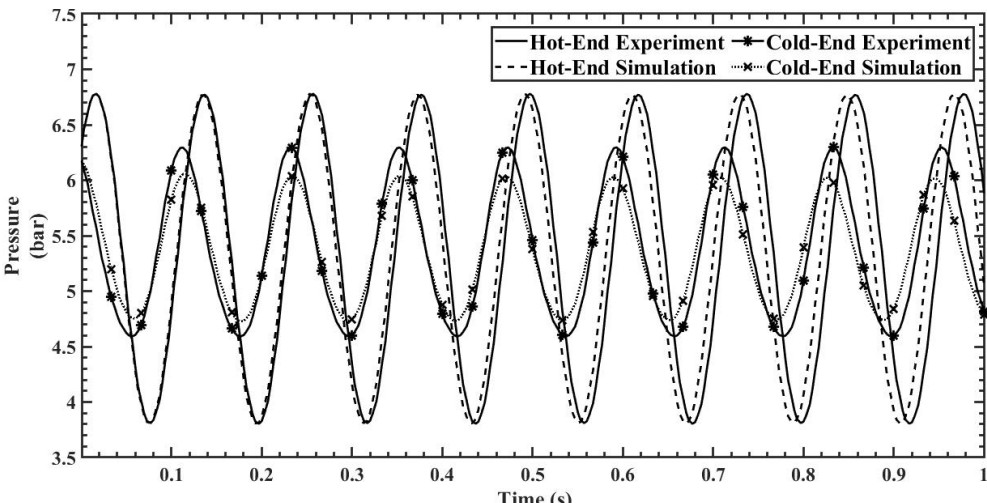

**Figure 17.** Hot- and cold-section pressure characteristics of a 200 mesh regenerator at 500 RPM.

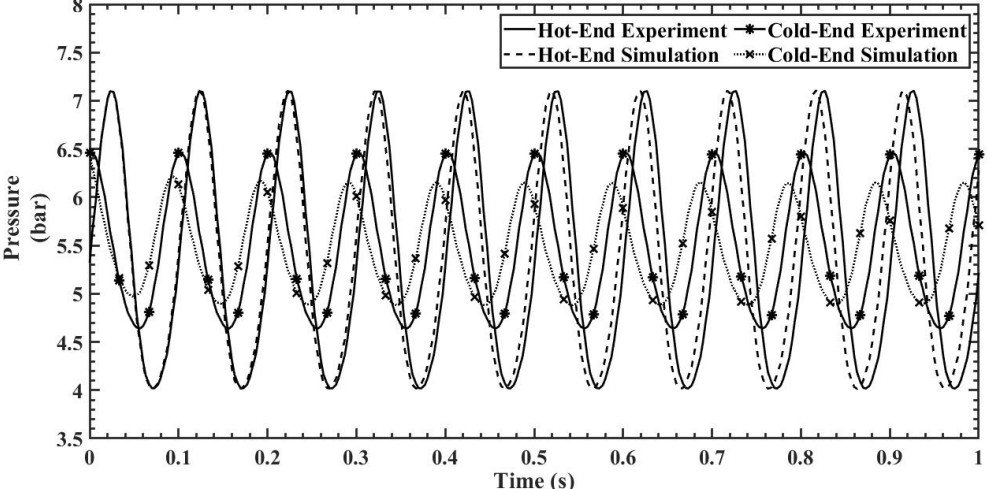

**Figure 18.** Hot- and cold-section pressure characteristics of a 200 mesh regenerator at 600 RPM.

For a 200 mesh regenerator with a fluid frequency of 200 RPM, the deviation of the maximum pressure from the experimental result is found to be 5.3%, approximately 1%

higher than the result obtained with a 300 mesh regenerator. It is found that for the 200 mesh regenerator, the error in numerical results is reduced as the operating frequency increases.

### 4.2. Temporal Pressure Drop Characteristics

The temporal pressure drop is calculated for all the cases from the pressure values obtained at both the hot and cold sections, as discussed in the previous section. The pressure drop (ΔP) is calculated by estimating the pressure difference between hot end and cold end for both experimental and numerical studies. For the sake of clarity and to avoid duplication, the pressure drop plots are shown only at two piston speeds for both regenerators since the temporal pressure variations are already shown in Figures 9–18. Figures 19 and 20 show the temporal pressure drop characteristics for the 300 mesh regenerator at piston speeds of 600 and 500 RPM, respectively. Similarly, Figures 21 and 22 show the temporal pressure drop characteristics for the 200 mesh regenerator at piston speeds of 600 and 500 RPM, respectively.

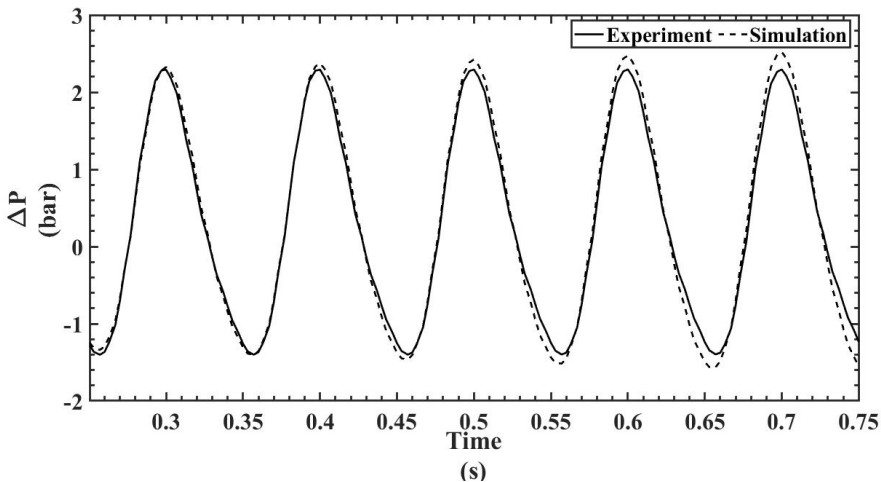

**Figure 19.** Temporal pressure drop for a 300 mesh regenerator at 600 RPM.

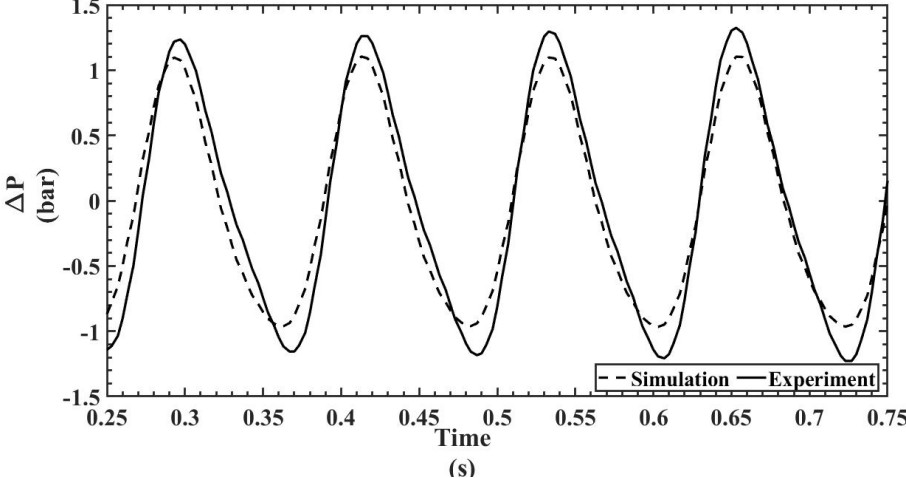

**Figure 20.** Temporal pressure drop for a 300 mesh regenerator at 500 RPM.

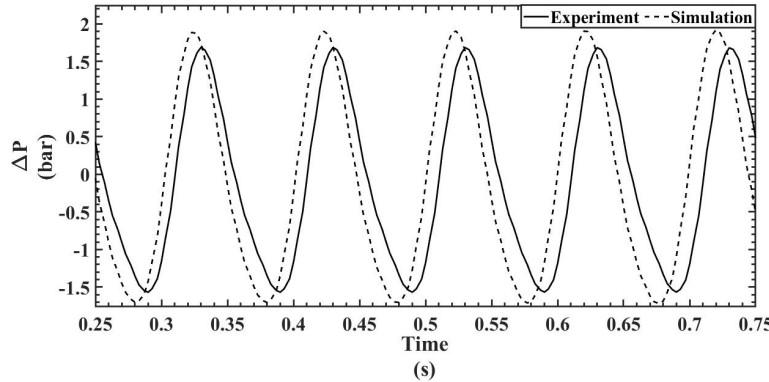

**Figure 21.** Temporal pressure drop for a 200 mesh regenerator at 600 RPM.

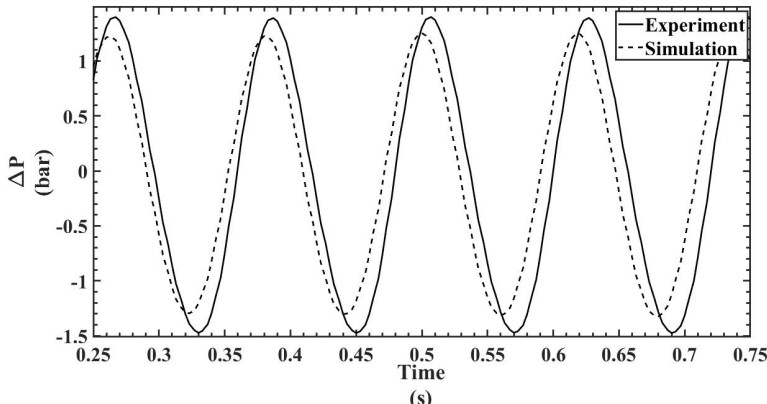

**Figure 22.** Temporal pressure drop for a 200 mesh regenerator at 500 RPM.

Comparing the pressure drop of 300 mesh and 200 mesh regenerators, it can be seen that the 300 mesh regenerator has a higher pressure drop. Therefore, it is clear that pressure drop is increased with an increase in porosity, and also, for both 300 and 200 mesh regenerators, the numerically obtained pressure drop is found to be reasonably in good agreement with the experimental results. The variation is more for 200 mesh regenerator cases. The maximum pressure drop values are also estimated for various Reynolds numbers based on the piston speed and plotted in Figure 23. It is observed that the maximum pressure drop increases with the bellow speed. However, the effect of porosity on pressure drop is clearly seen at higher bellow speeds, whereas it is less dominant at lower speeds.

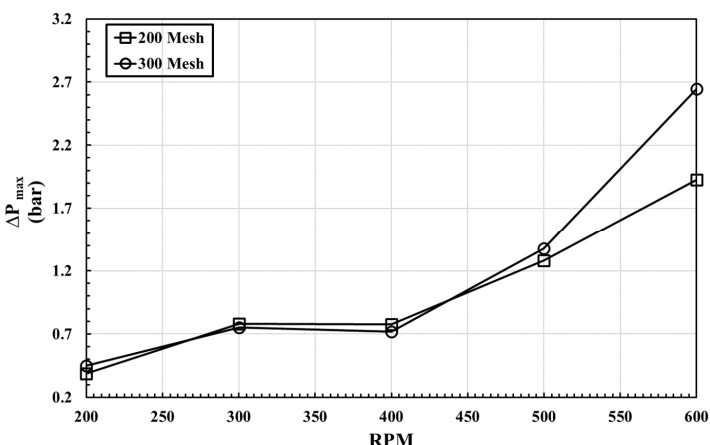

**Figure 23.** Variation of maximum pressure drop with piston speed for 200 and 300 mesh regenerators.

### 4.3. Pressure Variation Characteristics inside the Regenerator

Figures 24–27 include the pressure contours and centerline pressure variation plot along the axial flow direction for 500 RPM at different flow instants for 200 mesh and 300 mesh regenerators. It can be clearly seen from both plots that the pressure changes are bidirectional with time as expected since the regenerator will alternatively be experiencing compression and expansion phases with time. The plots also show that the instantaneous pressure variation in the regenerator is linear with the axial distance and depends on the porosity.

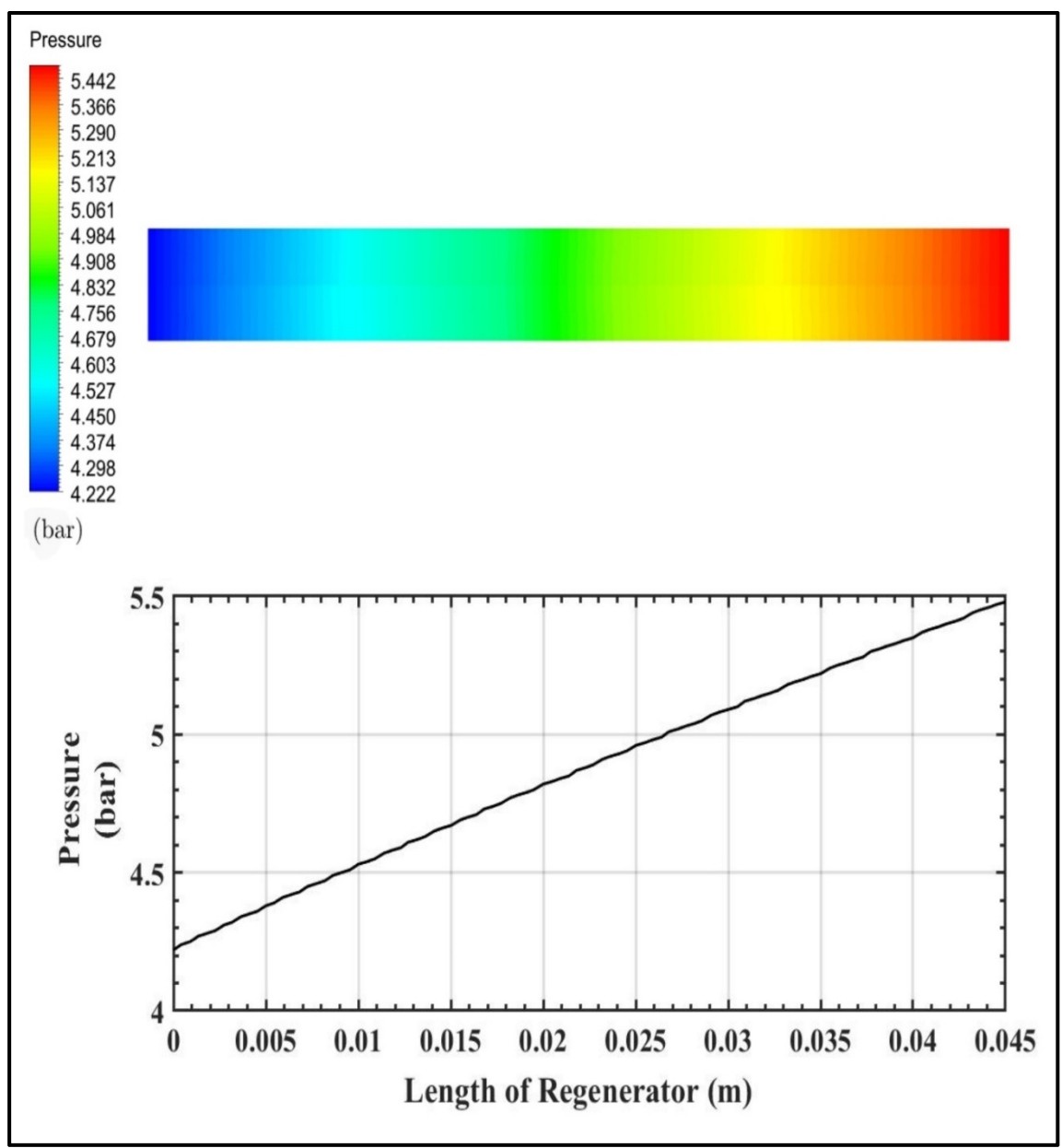

**Figure 24.** Variation of pressure along the length of the 200 mesh regenerator at 0.45 s.

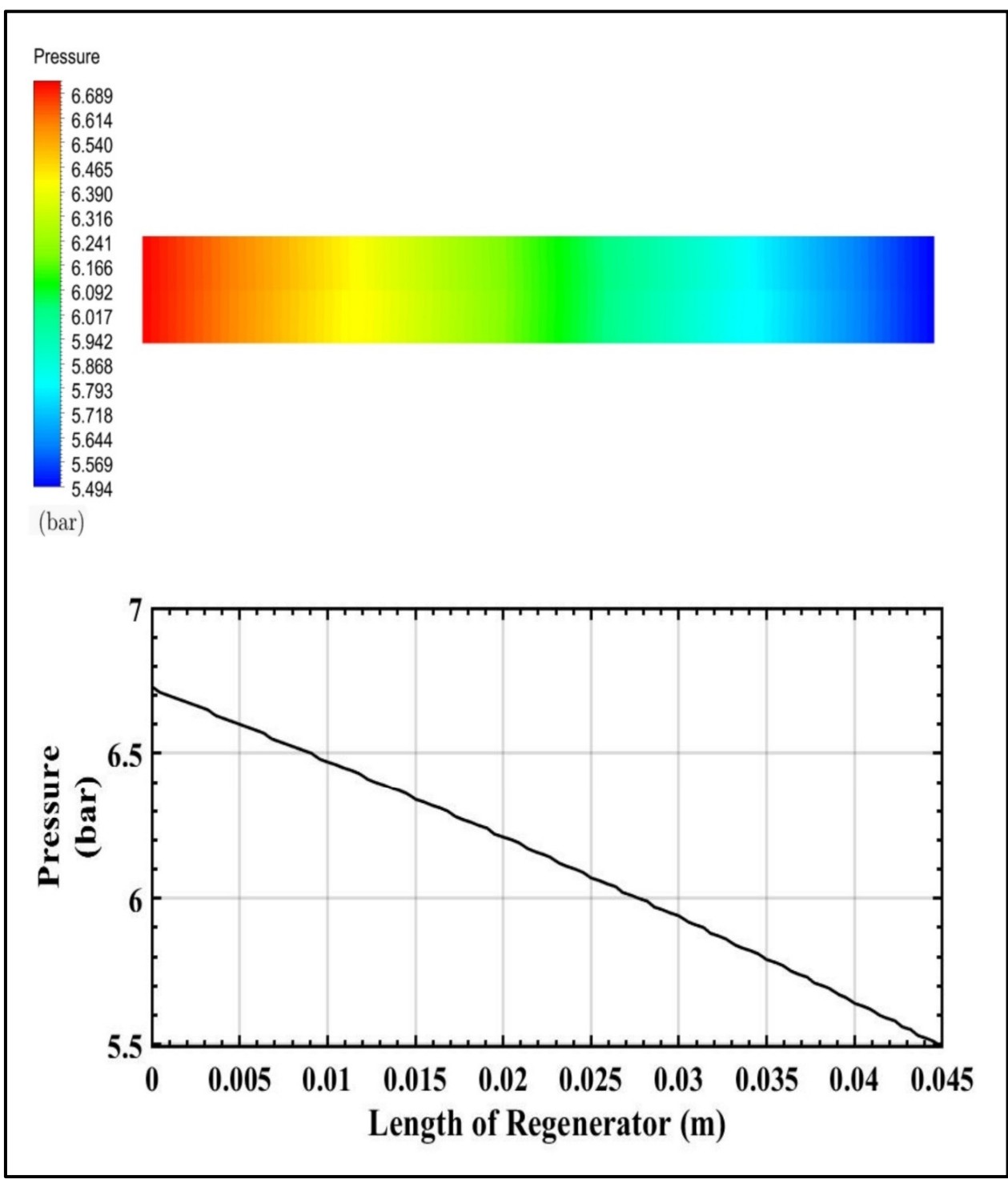

**Figure 25.** Variation of pressure along the length of the 200 mesh regenerator at 0.5 s.

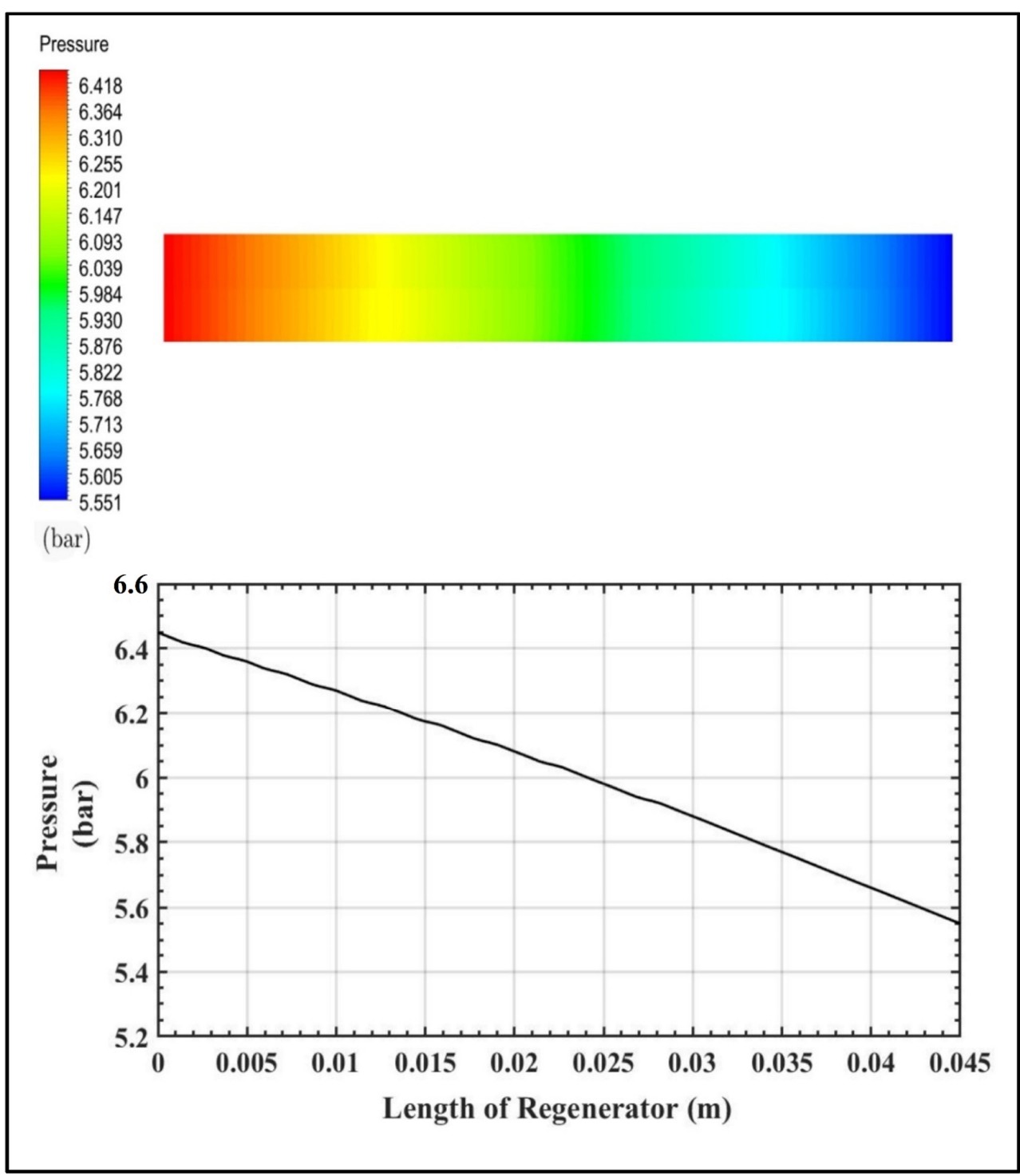

**Figure 26.** Variation of pressure along the length of the 300 mesh regenerator at 0.3 s.

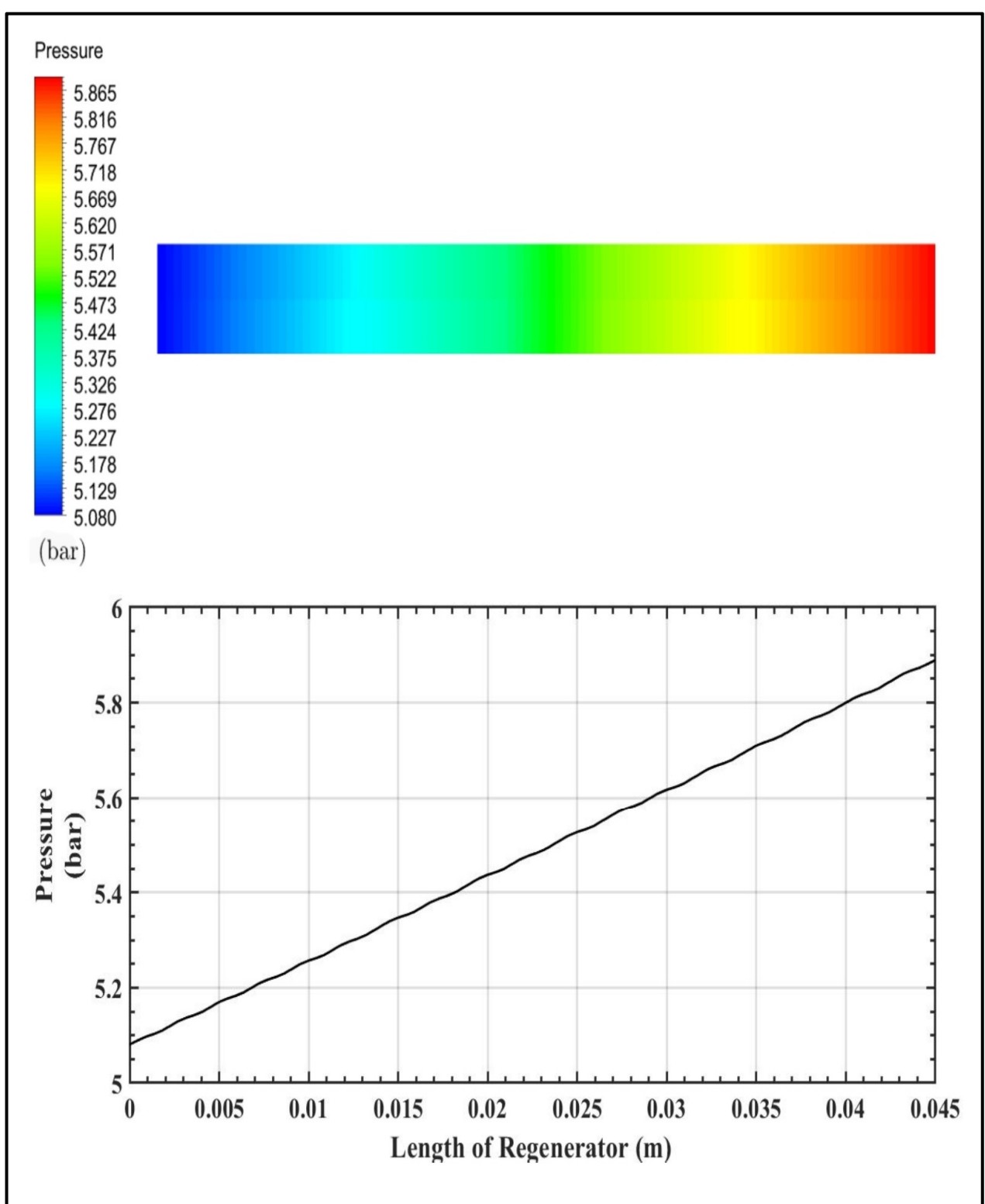

**Figure 27.** Variation of pressure along the length of the 300 mesh regenerator at 0.37 s.

### 4.4. Cause of Numerical Errors and Validity of Solution

The difference in pressure obtained in experiments and simulations can be due to the geometrical assumption that the model is 2D axisymmetric. Additionally, the leakages and the pressure increment in experiments due to bents and joints are neglected in the numerical study. The temporal lag in higher flow rates can be due to the application of the

sinusoidal function to define the motion of the piston mathematically. In real time, internal frictional effects affect the piston motion in the bellow, and those effects are not modelled in numerical results. Additionally, the application of the Ergun correlation is found to be valid for this problem only at higher frequencies, where convection plays a dominant role over diffusion. At lower flow rates or operating frequencies, it is essential to properly define the friction factor in order to acquire accurate results.

### 4.5. Friction Factor Characteristics

A friction factor correlation is developed for both 200 mesh and 300 mesh regenerators with the obtained viscous resistance and inertial resistance using cylindrical particle assumption. The friction factor correlation is obtained as follows:

Equation (9) shows the relation between pressure drop and friction factor based on the Darcy law.

$$f = \frac{\Delta P}{L} \frac{d_h}{\frac{1}{2}\rho u^2} \tag{9}$$

Equation (10) shows the relation between pressure drop and resistances based on the Ergun semiempirical correlation.

$$\frac{\Delta P}{L} = R_v \mu u + \frac{R_i}{2} \rho u^2 \tag{10}$$

By substituting Equation (10) in Equation (9),

$$f = \frac{2R_v d_h \mu}{\rho u} + R_i d_h \tag{11}$$

$$f = \frac{2R_v d_h^2 \nu}{u d_h} + R_i d_h \tag{12}$$

$$Re_{dh} = \frac{u d_h}{\nu} \tag{13}$$

By substituting Equation (13) in Equation (12), a general friction factor correlation is obtained based on viscous resistance and inertial resistance and shown in Equation (14).

$$f = \frac{2R_v d_h^2}{Re_{dh}} + R_i d_h \tag{14}$$

Substituting the known values of viscous resistance, inertial resistance, and hydraulic diameter, the friction factors for 200 mesh and 300 mesh regenerators are obtained and shown in Equations (15) and (16), respectively.

The friction factor for 200 mesh is given as:

$$f = \frac{252.47}{Re_{dh}} + 5.014 \tag{15}$$

The friction factor for 300 mesh is given as:

$$f = \frac{1273.07}{Re_{dh}} + 16.12 \tag{16}$$

Figure 28 shows the friction factor versus a Reynolds number graph. It can be noticed that the obtained friction factor trend matches those of previous studies [2,31–34,36]. It can be noticed that the friction factor is maximum at a low Reynolds number, and its significance is reduced with an increase in Reynolds number.

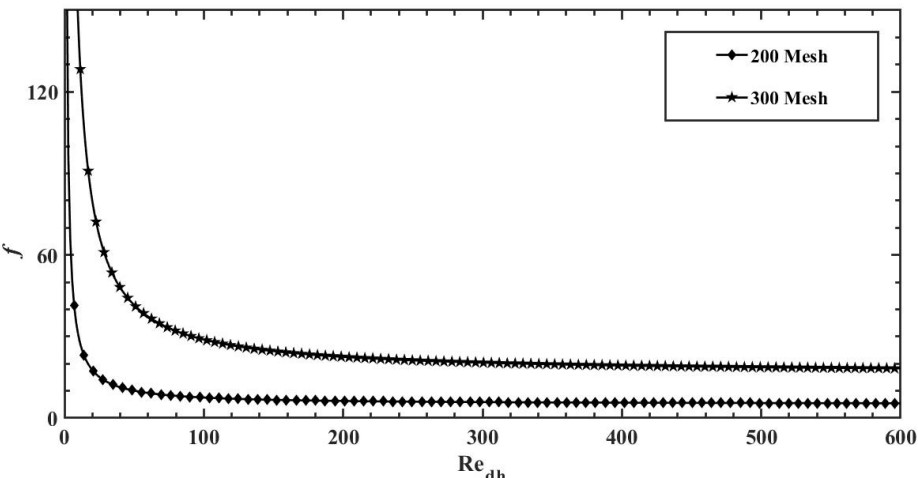

**Figure 28.** Friction factor calculated based on inertial and viscous Resistances.

## 5. Conclusions

A hydrodynamic study of an oscillating regenerator is performed both numerically and experimentally for two different mesh regenerators at different piston speeds. The oscillatory motion of the piston is implemented through a user-defined function using C programming libraries available in Ansys Fluent. The temporal pressure variations and pressure drops are computed for different piston speeds. It is found that the numerical results predict the oscillatory flow behaviors very reasonably well. The study reveals that the application of the Ergun correlation for a wire mesh does not have any effect with respect to the porosity of the regenerator. At lower operating frequencies, it is found that the error in numerical results is increased, and at higher flow rates, the numerical model is in good agreement with numerical results. At lower flow rates, due to inertial dominance, the Ergun correlation fails in predicting the proper pressure drop value. However, the Ergun correlation can be helpful in initial hydrodynamic studies on a regenerator at these conditions. For inertial dominant flows, it is essential to estimate and define the accurate value of the permeability and inertial coefficient. Simulation results clearly show the pressure variation along the regenerators and precite that the maximum pressure drop increases with piston speed for both regenerator meshes.

**Author Contributions:** Conceptualization, S.-W.K.; methodology, S.-W.K., S.S., K.J.B.; software, S.S., K.J.B.; validation, S.S., K.J.B.; formal analysis, S.-W.K., K.J.B., K.-L.C., K.-Y.L.; investigation, S.-W.K., K.J.B., S.S.; resources, S.-W.K., K.-L.C., K.-Y.L.; data curation, K.J.B.; writing—original draft preparation, S.-W.K., K.J.B.; writing—review and editing, S.-W.K., S.S., K.J.B.; visualization, K.J.B., K.-L.C., K.-Y.L.; supervision, S.-W.K., S.S.; project administration, S.-W.K., K.-L.C., K.-Y.L.; funding acquisition, S.-W.K. All authors have read and agreed to the published version of the manuscript.

**Funding:** This work was supported by the Ministry of Science and Technology, Taiwan, Republic of China, under contract number MOST 107-2622-E-006-005-CC2.

**Conflicts of Interest:** The authors declare no conflict of interest.

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
