# Peer review of "Correlations Based on Numerical Validation of Oscillating Flow Regenerator"

_processes, doi:10.3390/pr10071400_

Round 1

Reviewer 1 Report

This work conducted an experimental investigation on the pressure variation characteristics of oscillating flow in the regenerator of Stirling refrigerator, which is an interesting topic crucial to our understanding of working mechanism of the regenerative thermal engine and optimization of it. However, the reviewer didn’t find enough soundness throughout the authors’ work. A major revision is suggested to be made before this manuscript can be accepted. The followings are the suggestions or comments in details.

1. It is not clear the new idea in this research in terms of both experiments and simulations. In the introduction section, it is concluded the numerical studies “… is valid only within the experimental regime.” This should not be considered as a barrier of previous studies. If this work focuses on the accuracy of Ergun’s correlation and its uniform cylindrical particles assumption, comparisons of the CFD performance between the results using this model and those using other models should be presented in the analysis section.

2. The experiment section is lack of an error analysis. After obtaining the pressure wave along the timeline, the pressure drop along the axis from the experiments should be presented. Similar pressure waves don’t necessarily indicate good agreement with regards to pressure drop due to the propagation of error. Hence, an error analysis is necessary for the experimental results.

3. To the understanding of present reviewer, the instructive point of this type of research is the overall pressure drop characterized by friction factor as listed in Table 1. However, the present manuscript only includes the pressure drop simulation using Ergun's model. To validate this numerical model in analyzing the oscillating flow characteristics of regenerator, it is necessary to include the comparison with experimental results. The authors are also suggested to create a similar expression form for competing with the existing correlations, so as to illustrate the applicability and accuracy of the present model.

Reviewer 2 Report

o.k. I think the topic treated on the manuscript entitled “Correlation Based Numerical Validation of Oscillating Flow Regenerator” encoded as: processes-1790407 is very interesting the work was well driven and can be a nice contribution; nevertheless there some detail must be corrected in order to accept this manuscript for publication, next I will mention.

1.-The are only a few grammatical error as is shown in the attached file.

2.-Introduction begins on line 27 and finished on line 168, here authors explain in detail previous works done by other authors. In my personal opinion is too long, maybe can group works and reduce size.

3.- Figure 1. I suggest please make the fonts bigger.

4.- On figures 11,12 & 13 is necessary a (1) at the end of the horizontal axis in order to have all figures with homogeneity. The same on figure 20 at the end please place 0.75.

5.- about results on graphics I have the following questions.

5.1.-On figures 9 to 18 seems that the approaching between simulation and experiments was better for hot ends than cold ends for all cases on 200 and 300 mesh and for all RPMs calculated. Is there any reason related with numerical simulation?

5.2.- On figures 9 to 18 figures for curves of cold ends seem to be lightly run behind than hot ends, I am sure depends of the position on the regenerator and the evolution of the simulation, can you explain please.

5.3 On figures 9 to 18 seems that pressure drop is increased as the RPMs are increased, this fact is also confirmed on figure 23, can you describe the hydrodynamic reason please? 

5.4 On figure 13 there are only 3 curves (hot end simulation is absent) please correct including in this figure.

5.5. On the figures 9 to 18 the oscillations are increased as the RPMs are increased, the same behavior is observed for all curves, can you tell us about the reason; moreover, about this fact the period of every sinusoidal curve is reduced. As an example, on figure 16 for 400 RPMs the period is nearly 1,4 sec and on figure 18 it seems the value is 1.0 sec for 600 RPMs. Does it is important? and explain why please.

5.6 please can you explain how curves on figures 19 to 22 were obtained. I think as follow:

Experimental= Hot experimental – cold experimental

&

Simulated=hot simulated - cold simulated.

And the process was repeated according with every condition mentioned, please include a brief mention and equations.

I hope this review helps you to improve your work. I really think, this is a nice paper and will be accepted as soon as you make changes.

Round 2

Reviewer 1 Report

The suggestions for previous version have been well responded.